# Identification of Cancer Stem Cell Subpopulations in Head and Neck Metastatic Malignant Melanoma

**DOI:** 10.3390/cells9020324

**Published:** 2020-01-30

**Authors:** Vithushiya Yoganandarajah, Josie Patel, Bede van Schaijik, Nicholas Bockett, Helen D. Brasch, Erin Paterson, Dalice Sim, Paul F. Davis, Imogen M. Roth, Tinte Itinteang, Swee T. Tan

**Affiliations:** 1Gillies McIndoe Research Institute, Newtown, Wellington 6021, New Zealand; vithushiya@gmail.com (V.Y.); josie.patel@gmri.org.nz (J.P.); bedevs@gmail.com (B.v.S.); nick.bockett@gmri.org.nz (N.B.); helen.brasch@gmri.org.nz (H.D.B.); erin.paterson@gmri.org.nz (E.P.); paul.davis@gmri.org.nz (P.F.D.); imogen.roth@gmri.org.nz (I.M.R.); tinte01@yahoo.com (T.I.); 2Biostatistical Group/Dean’s Department, University of Otago, Wellington 6242, New Zealand; simdalice@gmail.com; 3Wellington Regional Plastic, Maxillofacial and Burns Unit, Hutt Hospital, Lower Hutt 5010, New Zealand; 4Department of Surgery, The University of Melbourne, Parkville, Victoria 3050, Australia

**Keywords:** malignant melanoma, head and neck cancer, cancer stem cell, induced pluripotent stem cell, melanoma metastasis

## Abstract

Cancer stem cells (CSCs) have been identified in many cancer types. This study identified and characterized CSCs in head and neck metastatic malignant melanoma (HNmMM) to regional lymph nodes using induced pluripotent stem cell (iPSC) markers. Immunohistochemical (IHC) staining performed on 20 HNmMM tissue samples demonstrated expression of iPSC markers OCT4, SOX2, KLF4, and c-MYC in all samples, while NANOG was expressed at low levels in two samples. Immunofluorescence (IF) staining demonstrated an OCT4+/SOX2+/KLF4+/c-MYC+ CSC subpopulation within the tumor nests (TNs) and another within the peritumoral stroma (PTS) of HNmMM tissues. IF also showed expression of NANOG by some OCT4+/SOX2+/KLF4+/c-MYC+ cells within the TNs in an HNmMM tissue sample that expressed NANOG on IHC staining. In situ hybridization (*n* = 6) and reverse-transcription quantitative polymerase chain reaction (*n* = 5) on the HNmMM samples confirmed expression of all five iPSC markers. Western blotting of primary cell lines derived from four of the 20 HNmMM tissue samples showed expression of SOX2, KLF4, and c-MYC but not OCT4 and NANOG, and three of these cell lines formed tumorspheres in vitro. We demonstrate the presence of two putative CSC subpopulations within HNmMM, which may be a novel therapeutic target in the treatment of this aggressive cancer.

## 1. Introduction

In 2018, 287,723 new cases of malignant melanoma (MM) were diagnosed worldwide with a record of 60,712 deaths [1]. MM accounts for 60–80% of deaths from all skin cancers globally with New Zealand and Australia having the highest incidence [2,3,4]. The 5-year survival of metastatic MM (mMM) is 5–19% depending on the site of the metastasis [2,5] with an overall median survival of 5–9 months [6,7].

Sentinel lymph node biopsy confers prognostic value and a potential survival advantage for patients with nodal metastases from intermediate-thickness MM who are treated with elective nodal dissection [8,9]. Surgical excision and adjuvant radiotherapy are the mainstay treatment for nodal mMM, with distant metastases requiring chemotherapy and/or immunotherapy [8,9].

MM commonly occurs in the head and neck region [10]. Despite intensive research, the prognosis for head and neck mMM (HNmMM) to regional lymph nodes remains poor with a 5-year survival rate of 17–22.6% [11,12]. Stage III and IV disease with metastases to regional lymph nodes as well as to distant organs, such as the lungs, liver, brain, and bones, are associated with 1-year survival rates of 7, 6–7, 6–9, and 2–4 months, respectively [13,14,15].

It has been proposed that the growth and spread of cancer is driven by a subpopulation of cancer cells known as cancer stem cells (CSCs) that are formed following acquisition of mutations in resident physiologically normal embryonic stem cells (ESCs) or progenitor cells [16]. These CSCs are capable of proliferation and self-renewal as well as multi-lineage differentiation into non-tumorigenic cells that are non-proliferative and lack self-renewability [17]. Thus, these CSCs operate in a hierarchical fashion similar to physiological stem cells and contribute to the heterogeneity of the tumor tissue. 

Reprogramming of these CSCs requires changes in gene expression that are directly modulated by transcription factors [18]. As such, octamer-binding transcription factor 4 (OCT4), homeobox protein NANOG, Kruppel-like factor 4 (KLF4), sex-determining region Y-box 2 (SOX2), and c-MYC are transcription factors regulating pluripotency and self-renewal of ESCs and have been shown to reprogram differentiated somatic cells, including melanocytes, into induced pluripotent stem cells (iPSCs) [19,20]. OCT4, NANOG, and c-MYC overexpression has been implicated in metastasis in many cancer types, including MM, in which cells expressing these markers have been shown to confer aggressive motility phenotypes, thus promoting invasiveness [21,22,23]. Interestingly, OCT4 has also been shown to drive dedifferentiation of MM cells into a CSC-like phenotype with increased tumorigenicity and metastatic capacity [24]. The pro-oncogenic role of KLF4 has been demonstrated in MM cell lines [25], while the tumor initiation and maintenance role of SOX2 has also been well-documented [19]. All or some of these iPSC markers have been used to identify CSCs in many cancer types including squamous cell carcinoma (SCC) affecting the oral tongue [26], lip [27], buccal mucosa [28], and skin [29]; breast cancer [30]; glioblastoma [31,32]; renal clear cell carcinoma [33]; acute myeloid leukemia [34]; primary MM [35]; metastatic MM (mMM) to the brain [15]; rectal cancer [36]; primary colon adenocarcinoma [37]; and metastatic colon adenocarcinoma to the liver [38].

Despite recent literature showing the presence of CSCs in many cancer types, there is a dearth of information characterizing the presence of CSCs within HNmMM using the iPSC markers. In this study, we aimed to identify and characterize CSCs using the iPSC markers OCT4, NANOG, SOX2, KLF4, and c-MYC in HNmMM to the parotid and/or neck nodes at both the transcript and protein levels. Here, we demonstrate the presence of an OCT4+/SOX2+/KLF4+/c-MYC+ CSC subpopulation within the tumor nests (TNs) and another within the peritumoral stroma (PTS), in this tumor.

## 2. Materials and Methods

### 2.1. HNmMM Tissue Samples

Tissue samples of HNmMM to the parotid and/or neck nodes of 16 male and 4 female patients aged 47–103 (median, 75) years, with a known primary melanoma in the head and neck region (Appendix A), were sourced from the Gillies McIndoe Research Institute Tissue Bank, and used in a study conducted in accordance with the Declaration of Helsinki, with the protocol being approved by the Central Health and Disabilities Ethics Committee (Ref. 12/CEN/74) with written informed consent from all participants prior to their participation.

### 2.2. HNmMM-Derived Primary Cell Lines

HNmMM-derived primary cell lines were established from four fresh surgically excised HNmMM tissue samples from the original cohort of 20 patients by culturing them within a Matrigel explant prior to extraction of the cells following abundant growth in 24-well plates (Raylab, Auckland, New Zealand). The extracted cells were cultured in a cell culture medium consisting of DMEM (1X) and GlutaMAX-1 (cat#10569-010, Gibco, Rockford, IL, USA) supplemented with 10% fetal bovine serum (cat# 10091148, Gibco), 5% mTeSR 1 Complete medium (cat#85850, STEMCELL Technologies, Vancouver, BC, Canada), 1% penicillin-streptomycin (cat#15140122, Gibco), and 0.2% gentamycin/amphotericin B (cat#R015-10, Gibco). All cultures were maintained in a humidified incubator at 37 °C with 5% CO_2_. All primary cell lines used for the experiments were at passages 4–5 (RT-qPCR and WB) or 8–9 (tumorsphere formation).

### 2.3. Histochemical, Immunohistochemical, and Immunofluorescence Staining

Hematoxylin and eosin (H&E) staining and Melan-A staining were carried out on 4 μm thick formalin-fixed paraffin-embedded (FFPE) sections of all 20 HNmMM tissue samples to confirm the presence and diagnosis of the tumor. Tumors that were negative for Melan-A were stained for S100 protein, an alternate confirmatory immunohistochemical (IHC) stain for melanoma. IHC staining of these consecutive sections was then undertaken using the Leica BOND RX™ auto-stainer (Leica Biosystems, Nussloch, Germany). Staining for Melan-A (ready-to-use; cat#PA0233, Leica Biosystems, Newcastle-upon-Tyne, UK), S100 (1:200; cat#330M, Cell Marque, Rocklin, CA, USA), OCT4 (1:30; cat#MRQ-10, Cell Marque), NANOG (1:200; cat#EP225, Cell Marque), SOX2 (1:500; cat#PA1-094, ThermoFisher Scientific, Rockford, IL, USA), KLF4 (1:100; cat#NBP2-24749, Novus Biologicals LLC, Littleton, CO, USA), and c-MYC (1:1000; cat#9E10, Abcam, Cambridge, UK) was performed with 3,3-diaminobenzidine as the chromogen, and the antibodies were diluted with BOND^TM^ primary antibody diluent (Leica). All IHC-stained slides were mounted in Surgipath Micromount mounting medium (cat#381017322, Leica).

Immunofluorescence (IF) staining was performed on consecutive sections from the same tissue blocks used for IHC staining on two HNmMM tissue samples from the original cohort of 20 patients. This was performed to demonstrate colocalization of two markers using a combination of VectaFluor Excel anti-rabbit 594 (ready-to use; cat#VEDK-1594, Vector Laboratories, Burlingame, CA, USA) and Alexa Fluor anti-mouse 488 (1:500; cat#A21202, Life Technologies) to detect combinations that included OCT4 and c-MYC. Dual-staining of Melan-A with NANOG, SOX2, KLF4, and c- MYC was also performed on the same two HNmMM tissue samples to investigate expression of these markers on the Melan-A+ cells within HNmMM tissue samples. A Melan-A/OCT4 combination was not done due to species incompatibility of the primary antibodies available. VectaFluor Excel anti-mouse (ready-to-use; cat#VEDK2488, Vector Laboratories) and Alexa Fluor anti-rabbit 594 (1:500; cat#A21207, Life Technologies) were used to detect combinations that included NANOG, SOX2, and KLF4.

Human tissues used for positive controls for the primary antibodies were seminoma for OCT4 and NANOG, normal skin for SOX2, breast carcinoma for KLF4, and normal colon for c-MYC. A negative antibody control was carried out on one randomly selected HNmMM sample per antibody staining run with staining using either a mouse (ready-to-use; cat#IR750, Dako, Carpinteria, CA, USA) or rabbit (ready-to-use; cat#IR600, Dako) primary antibody isotype-matched control for the IHC staining or a combination for the IF staining.

### 2.4. In situ Hybridization

In situ hybridization (ISH) was carried out on 4 μm thick FFPE consecutive sections from the same tissue blocks used for IHC and IF staining on six randomly selected HNmMM samples from the original cohort of 20 patients using OCT4 (cat#592868), NANOG (cat#604498), SOX2 (cat#477651), KLF4 (cat#457468), and c-MYC (cat#311768-C2) probes (Advanced Cell Diagnostics, Newark, CA, USA) and detected using RNAscope 2.5LS Reagent Brown Kit (cat#322100, Advanced Cell Diagnostics). Positive control tissue sections were used as for IHC sections. Negative controls were demonstrated on sections of HNmMM using a probe for DapB (cat#3120358, Advanced Cell Diagnostics).

### 2.5. Image Analyses and Cell Counting

IHC and ISH stained slides were visualized and imaged using an Olympus BX53 microscope fitted with an Olympus SC100 digital camera (Olympus, Tokyo, Japan) and processed with the CellSens 2.0 Software (Olympus).

Representative areas of iPSC marker distributions were selected, captured, and counted (both within the TNs and the PTS). For IHC staining, three representative fields of each of the 19 sections were selected and counted at 400× magnification, and for ISH, six fields of view were selected, counted at 1000× magnification using the Cell Counter program on ImageJ, and presented as an average proportion of the total number of marker-positive cells in the total number of fields. A cell was regarded as positively stained for OCT4, SOX2, NANOG, KLF4, and c-MYC, if it resembled the positive control for the marker and the staining was present in either the nucleus or cytoplasm. Cells were distinguished from one another by the presence of their nuclei. All positively stained cells were counted and the proportions of positively stained cells out of the total number of cells within the field of view were then calculated and averaged across the three fields of view that had a minimum of 100 cells per field for each of the 19 cases. Cell counting on ISH-stained slides was performed in the same manner, except the images were taken at 1000× magnification with each view having a minimum of 10 cells for each of the six cases. One sample out of the original cohort of 20 patients was excluded from counting due to potential confounding effect owing to dense melanin pigmentation.

IF-stained images were visualized and captured using an Olympus FV1200 biological confocal laser-scanning microscope (Olympus) and subsequently processed with the cellSens Dimension 1.11 software using 2D deconvolution algorithms (Olympus).

### 2.6. Reverse Transcription Quantitative Polymerase Chain Reaction

Total RNA was isolated from five snap-frozen HNmMM tissue samples and four HNmMM-derived primary cell lines from the original cohort of 20 patients. Approximately 20 mg of tissue was homogenized using the Omni Tissue Homogenizer (Omni TH, Omni International, Kennesaw, GA, USA) and then prepared using the RNeasy Mini Kit (cat#74104, Qiagen, Hilden, Germany). Frozen cell pellets of approximately 5 × 10^5^ viable cells were prepared using the RNeasy Micro Kit (cat#74004, Qiagen). A DNase digest step was included to remove potentially contaminating genomic DNA (cat#79254, Qiagen). The quantity and quality of the RNA were determined using a NanoDrop 2000 Spectrophotometer (ThermoFisher Scientific). Transcript expression was assessed using the Rotor-Gene Q (Qiagen) and the Rotor-Gene Multiplex RT-PCR Kit (cat#204974, Qiagen). The primer probes used were OCT4 (Hs03005111_g1), NANOG (Hs02387400_g1), SOX2 (Hs01053049_s1), KLF4 (Hs00358836_m1), and c-MYC (Hs00153408_m1; cat#4331182) (ThermoFisher Scientific). Gene expression was normalized against the housekeepers GAPDH (Hs99999905_m1) and PSMB4 (Hs00160598_m1; cat#4331182, ThermoFisher Scientific). Universal human reference RNA (UHR; cat#CLT636690, Takara Bio Mountain View, CA, USA) was used as a control for the ∆∆Ct analysis, with expression presented as fold change relative to UHR. The qPCR UHR is a mixture of total RNAs pooled from multiple adult human tissues chosen to depict a wide range of expressed genes. It has been established as a reliable reference standard for the accurate and reproducible comparison of gene expression data using RT-qPCR [39,40]. The fold-change cut-off point was set at 2.0 for up-regulated and 0.5 for down-regulated genes.

Positive controls were demonstrated on NTERA2 cell lines with RNase-free water. Negative controls were used to confirm no contamination, and reverse-transcriptase negative controls were included for those primers that may detect genomic DNA. End-point PCR amplification product specificity was checked using 2% agarose gel (cat#G402002, ThermoFisher Scientific) electrophoresis using the eGel equipment (cat#G6400, ThermoFisher Scientific) and subsequent imaging using the ChemiDoc MP (Bio-Rad Laboratories, Hercules, CA, USA).

### 2.7. Western Blotting

Total protein extracts from four HNmMM-derived primary cell lines were resolved by 4%–12% one-dimensional polyacrylamide gel electrophoresis (ThermoFisher Scientific Carlsbad, CA, USA) at 20 μg total protein per sample and transferred to polyvinylidene difluoride membranes (ThermoFisher Scientific, Kiryat Schmona, Israel). The membranes were then probed with the following primary antibodies: OCT4 (1:1000; cat#ab109183, Abcam), NANOG (1:500; cat#ab109250, Abcam), SOX2 (1:500; cat#48-1400, ThermoFisher Scientific, Rockford, IL, USA), KLF4 (1:1000; cat#NBP2-24749, Novus Biologicals, Centennial, CO, USA), c-MYC (1:1000 cat#ab32072, Abcam), and α-tubulin (1:1000; cat#62204, ThermoFisher Scientific, Carlsbad, CA, USA), followed by incubation with an appropriate secondary antibody, either goat anti-rabbit, horseradish peroxidase (HRP) conjugate (1:1000; cat#ab6721, Abcam) for the five iPSC markers or mouse IgGĸ binding protein, HRP conjugate (1:2000; cat#sc516102, Santa Cruz Biotechnology, Dallas, TX, USA) for α-tubulin. HRP-conjugated secondary antibody detection and visualization were performed using Clarity Western enhanced chemiluminescence substrate (Bio-Rad Laboratories) and a ChemiDoc MP imaging System (Bio-Rad Laboratories).

Image Lab 6.0 (Bio-Rad Laboratories) was used to perform densitometric quantification of OCT4, NANOG, SOX2, KLF4, and c-MYC expression, relative to α-tubulin. Single bands that corresponded with the expected size of the respective proteins were used for the analysis.

### 2.8. In-vitro Tumorsphere Formation Assays

Tumorsphere suspension cultures were developed using four HNmMM-derived primary cell lines. Briefly, 2.5 × 10^5^ live cells from adherent cultures at passages 8–9 were seeded in 25 mL StemXVivo serum-free tumorsphere media (cat#CCM012, R&D Systems, Minneapolis, MN, USA), supplemented as per the manufacturer’s protocol in T75 Nunclon^TM^ Sphera^TM^ EasYFlasks (cat#174952, ThermoFisher Scientific), and cells were maintained in a 5% CO_2_ incubator at 37°C for up to 10 days or until initial signs of dark centers appearing on the spheres indicating necrosis were observed. Flasks were fed every 3–4 days with the addition of 12.5 mL of media. Cultures were observed under an Olympus CKX53 Microscope (Tokyo, Japan) daily from day three post seeding. Three fields of view were captured for each cell line, and from each field of view, five representative tumorspheres were identified. Two measurements per sphere-like structure were performed to establish an average sphere size per flask. If the average measurements of the spheres were greater than 50 µm in size and more than 50% of spheres measured per field of view were over 50 µm, the culture was classified as positive for tumorspheres [41].

### 2.9. Statistical Analysis

Statistical analyses for comparison of the iPSC markers investigated by IHC staining and ISH of the HNmMM tissue samples were undertaken. Two types of analyses were done: a) an Analysis of Variance (ANOVA) using the percentage of marker-positive cells as the dependent variable and the iPSC markers and replicates as independent variables, ignoring the fact that the samples came from the same patient and the replicates from the same sample; hence, they are correlated; b) to accommodate these correlations, Generalized Estimating Equations [42] with a linear model that required minimal distributional assumptions were used to test whether the mean levels of the five iPSC markers were different. Given the multiple comparisons being made, post hoc analysis with Bonferroni corrections were performed.

## 3. Results

### 3.1. OCT4, NANOG, SOX2, KLF4, and c-MYC Proteins are Expressed in HNmMM Tissue Samples

In order to confirm the presence of HNmMM, the slides were stained with H&E that showed TNs, (arrows) surrounded by PTS (arrowheads) in all 20 tissue samples (Figure 1A). In order to determine the presence of the five iPSC markers, the HNmMM tissue sections underwent IHC staining. This showed nuclear and cytoplasmic expression of OCT4 in the cells within the TNs (Figure 1B, arrows) and the PTS (Figure 1B, arrowheads). SOX2 (Figure 1C), KLF4 (Figure 1D), and c-MYC (Figure 1E) were present in all 20 tissue samples while NANOG showed weak cytoplasmic staining in only two of the 20 samples (a representative image of the two is shown in Figure 1F). SOX2 was localized predominantly to the nucleus and occasionally the cytoplasm of the cells within the TNs (Figure 1C, arrows) and the cells within the PTS (Figure 1C, arrowheads). Ubiquitous cytoplasmic staining of KLF4 was found in the cells within the TNs (Figure 1D, arrows) and to a lesser extent in the cells within the PTS (Figure 1D, arrowheads), while c-MYC showed focal nuclear staining and moderate cytoplasmic staining of the cells within the TNs (Figure 1E, arrows) and the cells within the PTS (Figure 1E, arrowheads). In most of the samples, NANOG was not detected within the TNs. Appendix A shows the same IHC staining of iPSC markers with the same expression patterns in HNmMM tissues, as shown in Figure 1, at a lower magnification to better illustrate the demarcation between TNs and PTS.

Expected staining patterns were demonstrated in the human positive control tissues: seminoma for OCT4 (Appendix A) and NANOG (Appendix A), normal skin for SOX2 (Appendix A), breast carcinoma for KLF4 (Appendix A), and normal colon for c-MYC (Appendix A). The isotype-matched antibodies provided appropriate negative controls (Appendix A).

In order to compare the protein expression of different iPSC markers, we performed cell counting analysis of 19 IHC slides (one case was excluded due to dense melanin pigmentation, refer to Methods) of HNmMM, where all marker-positive cells in both the TNs and PTS were counted and recorded as a proportion of the total number of cells in the field of view. When comparing the total proportion of positively stained cells within the TNs and PTS for each marker, post-hoc statistical analysis demonstrated a hierarchy of expression of these markers with increasing abundance as follows: NANOG < OCT4 < KLF4 < c-MYC < SOX2 (Figure 2). All comparisons were highly statistically significant between markers (*p* < 0.0005) except for the comparisons between NANOG and OCT4, which was significant (*p* < 0.05), and between c-MYC and SOX2, which was not significant.

### 3.2. Subpopulations of CSCs Expressing OCT4, NANOG, SOX2, KLF4, and c-MYC are Present in HNmMM Tissue Samples

To investigate localization of two iPSC markers simultaneously, IF staining was performed on two representative HNmMM tissue samples. IF staining demonstrated expression of OCT4 (Figure 3A–C, green), SOX2 (Figure 3B,E, red), KLF4 (Figure 3C,F, red), and c-MYC (Figure 3D–F, green) by the cells within the TNs (thick arrows) and the PTS (arrowheads). NANOG (Figure 3A,D and inset, red) was present in one sample that showed NANOG expression on IHC staining and was absent in the other sample that did not show NANOG expression on IHC staining. OCT4 was expressed on the NANOG+ (Figure 3A and inset, red), the SOX2+ (Figure 3B and inset, red), and the KLF4+ (Figure 3C and inset, red) cells within the TNs and the PTS. c-MYC was also expressed by the NANOG+ (Figure 3D and inset, red), the SOX2+ (Figure 3E and inset, red), and the KLF4+ (Figure 3F and inset, red) cells within the TNs and the PTS. Interestingly, some c-MYC+ (Figure 3D and inset, green) cells within the TNs were NANOG- (Figure 3D, thin arrows). Magnified figure insets have been provided to show enlarged views of the corresponding images. IF dual-staining HNmMM tissue samples for the melanoma marker Melan-A and induced pluripotent stem cell markers showing expression of NANOG (**A**, red), SOX 2 (**B**, red) SOX2 (**C**, red), KLF4 (**C**, red), and c-MYC (**D**, red) on some Melan-A+ (**A**–**D,** green) cells within the tumor nests and some cells within the peritumoral stroma (Appendix A). A Melan-A/OCT4 combination was not done due to species incompatibility of the primary antibodies available. Minimal staining was present on the negative controls, confirming the specificity of the primary antibodies used (Appendix A).

Images of individual stains of the merged images presented in Figure 3 are provided in Appendix A. Minimal to no staining was present on the negative controls (Appendix A), confirming the specificity of the primary antibodies used.

### 3.3. OCT4, NANOG, SOX2, KLF4, and c-MYC mRNA Transcripts are Expressed in HNmMM Tissue Samples and Hnmmm-Derived Primary Cell Lines

In order to determine the transcript expression of the iPSC markers, in-situ hybridization (ISH) was performed on six HNmMM tissue samples. ISH demonstrated both nuclear and cytoplasmic presence of mRNA for OCT4 (Figure 4A), NANOG (Figure 4B), SOX2 (Figure 4C), KLF4 (Figure 4D) and c-MYC (Figure 4E) in the TNs in all six HNmMM tissue samples. All iPSC markers except NANOG were also expressed in the PTS. The transcript abundance of SOX2 (Figure 4C) and c-MYC (Figure 4E) is in line with the relatively increased expression of these two markers observed at the protein level in cell counting of IHC sections (Figure 2).

Positive controls demonstrated the expected staining pattern for OCT4 (Appendix A) and NANOG (Appendix A) in seminoma, SOX2 (Appendix A) in normal skin, KLF4 (Appendix A) in breast carcinoma, and c-MYC (Appendix A) in normal colon. The negative control (Appendix A) showed no staining, indicating no background activity.

In order to compare the transcript expression of different iPSC markers, we performed cell counting analysis of six ISH stained slides of HNmMM. When comparing the total proportion of positively stained cells within the TNs and the PTS for each marker, post hoc statistical analysis demonstrated a hierarchy of expression of these markers with increasing abundance: KLF4 < SOX2 < OCT4 < NANOG < c-MYC (Figure 5). Most comparisons were highly statistically significant between markers (*p* < 0.0005) except for the comparison between KLF4 and OCT4, which was statistically significant at *p* < 0.05. Comparisons between KLF4 and SOX2 and between SOX2 and OCT4 were not statistically significant.

In order to confirm the transcript expression of iPSC markers, we performed reverse transcription quantitative polymerase chain reaction (RT-qPCR) analysis of five snap-frozen HNmMM tissue samples and four HNmMM-derived primary cell lines. RT-qPCR analysis of both HNmMM tissue samples (Figure 6A) and HNmMM-derived primary cell lines (Figure 6B) demonstrated expression of all five iPSC markers, OCT4, NANOG, SOX2, KLF4, and c-MYC. Two of the five tissue samples showed an up-regulation in c-MYC (over 2-fold change). KLF4, NANOG, and SOX2 do not show a biologically significant change in mRNA expression relative to UHR, whereas OCT4 appears to be down-regulated. OCT4, NANOG, and SOX2 were all down-regulated in HNmMM-derived primary cell lines, compared to UHR. There was no biologically significant change in the mRNA expression of c-MYC or KLF4. Specific amplification of the products was demonstrated by electrophoresis of qPCR products on 2% agarose gels (Appendix A). The expected size amplicons were observed, and no products were observed in the no template control (NTC) reactions (Appendix A).

### 3.4. SOX2, KLF4, and c-MYC Proteins are Expressed in the HNmMM-Derived Primary Cell Lines

Western blot (WB) analysis of total protein extracts from the four HNmMM-derived primary cell lines showed that OCT4 (Figure 7A, ~39kDa) and NANOG (Figure 7B, ~40kDa) proteins were below detectable levels. SOX2 was detected in one of the four cell lines at ~38kDa, and faint bands were seen in a further two cell lines (Figure 7C). A band of ~40kDa was detected in the NTERA2 cell extract, the difference in size is likely to be due to post-translational modifications. The specificity of this antibody was confirmed using an alternative antibody for SOX2 from a different manufacturer and host species, revealing identical banding patterns (Appendix A). KLF4 showed specific bands at ~60kDa and, a smaller and likely non-specific band, at ~27kDa (Figure 7D). c-MYC showed a band at ~57kDa in the HNmMM cell extracts as well as in the positive control and a smaller ~40kDa band in two of the HNmMM cell extracts (Figure 7E) which could represent MYC-Nick [43]. α-Tubulin staining confirmed approximately equal total protein loading for the four cell line samples (Figure 7F, ~50kDa).

### 3.5. HNmMM-Derived Primary Cell Lines underwent Tumorsphere Formation

Tumorsphere cultivation is generally regarded as a functional assay of the self-renewal property of CSCs and to enrich these cells from bulk tumor cells. Cells that form tumorspheres are generally recognized as pluripotent CSCs that display the iPSC factors that imbue the tumor with self-renewal and tumorigenic properties [44,45]. In order to determine the sphere-forming ability of the CSCs within the HNmMM-derived primary cell lines, we cultured them in tumorsphere media. Three out of four of the cell lines cultured demonstrated tumorsphere formation (Figure 8A–C, high magnification; Appendix A, low magnification) that met our threshold (spheres that were greater than 50 µm in size, and more than 50% of spheres measured per field of view were over 50 µm [41]), while one cell line formed spheres but did not meet our threshold (less than 50% of spheres measured per field were over 50µm) (Figure 8D, high magnification; Appendix A, low magnification) in tumorsphere media, providing preliminary evidence that some of the cultured cells may be CSCs. 

Taken together, our results show that OCT4, SOX2, KLF4, and c-MYC proteins and transcripts are present within the HNmMM tissue samples and HNmMM-derived primary cell lines. Two out of the 20 HNmMM samples also expressed NANOG. An OCT4+/SOX2+/KLF4+/c-MYC+ subpopulation emerged in two distinct locations within HNmMM: one within the TNs and the other within the PTS. Interestingly, the HNmMM sample that was NANOG^+^ on IHC staining contained OCT4+/SOX2+/KLF4+/c-MYC+ cells within the TNs on IF staining that were NANOG+ as well as some that were NANOG-. Cell counting analysis of IHC and ISH revealed variable expression of each marker across the tissues. Three out of four HNmMM-derived primary cell lines formed tumorspheres in vitro, demonstrating the functionality of these cells as CSCs.

## 4. Discussion

The CSC concept proposes that cancer arises from a pool of cancer stem cells that can give rise to all the cell types within a tumor capable of recreating the heterogeneous phenotype of a tumor tissue, including producing more tumorigenic CSCs and non-tumorigenic cancer cells. Classically, CSC subpopulations have been defined by their ability to express various markers that have different phenotypic and functional characteristics, even within the same tumor [46]. In this study, we demonstrate the presence of an OCT4+/SOX2+/KLF4+/c-MYC+ CSC subpopulation within the TNs, and another within the PTS in the 20 HNmMM tissue samples analyzed. NANOG expression was detected at low levels, both qualitatively using IHC (Figure 1C) and IF (Figure 3A,D) staining, and quantitatively using cell counting (Figure 2) in two of the 20 samples analyzed. Interestingly, IF staining performed on one of the HNmMM samples that expressed NANOG on IHC staining showed that some of the OCT4+/SOX2+/KLF4+/c-MYC+ CSCs within the TNs also expressed NANOG.

OCT4 plays a major role in sustaining stem cell self-renewal [47] and causing dedifferentiation of melanoma cells to acquire a CSC phenotype [24]. Our findings are consistent with the expression of OCT4 previously observed in MM, which has been shown to confer chemoresistance and increased motility and invasiveness of melanoma cells [21]. While OCT4 was present in the TNs and the PTS in all 20 HNmMM tissue samples, it was not detected in the four HNmMM-derived primary cell lines examined by WB. This may be attributed to the relatively low levels of the protein in the cell lines compared to the tissues and/or sampling bias. With respect to the observed subcellular localization of OCT4, previous studies have shown that its localization is insignificant in relation to tumorigenesis as long as nuclear processing is maintained [48].

NANOG protein expression was not detected in 18 out of 20 HNmMM tissue samples by IHC staining (not shown) and in both HNmMM-derived primary cell lines by WB. The low cytoplasmic expression by cells within the TNs of only two of the 20 IHC-stained samples along with sparse expression on the c-MYC+ cells within the TNs demonstrated by IF staining reflects the dispensability of NANOG in the generation of iPSCs [49,50] and, perhaps, its dispensability in CSC formation. Interestingly, NANOG was detected by ISH within both the nucleus and cytoplasm of the cells within the TNs and by RT-qPCR. Possible reasons for this include a delay in translation of NANOG, perhaps due to the transitioning nature of these CSCs, perhaps following differentiation as the maturation, exportation, and translation of mRNA is a time-intensive process [51,52]. While known to maintain pluripotency, the function of subcellular localization of NANOG in iPSCs remains unclear [48].

SOX2 expression observed in both the nucleus and the cytoplasm of the cells (Figure 1C and Figure 3B) within the TNs and the PTS of HNmMM tissue samples and its presence in HNmMM-derived primary cell lines may be associated with CSC maintenance and tumor growth [19].

Interestingly, WB analysis for both SOX2 and KLF4 showed smaller bands at 27kDa that appeared to be non-specific as evidenced by their presence in IgG isotype-matched controls. It is likely that these could also represent degradation products of the proteins given their abundant expression in the positive control. The two bands representing SOX2 at 39–43 kDa in the NTERA2 cells are likely to be due to post-translational modifications occurring within the cell line [53,54]. Our findings show both a nuclear and cytoplasmic localization of SOX2, and this is supported in the literature where both a nuclear and cytoplasmic role for SOX2 has been described in CSC formation [48].

Abundant c-MYC expression in MM is associated with immune evasion and tumor invasiveness [55], thus its association with poor prognosis and survival [23]. We have observed consistently high cytoplasmic and moderate nuclear expression of c-MYC in the cells within the TNs and the PTS of HNmMM by IHC staining and ISH. This is consistent with c-MYC overexpression observed in most other human malignancies including ocular MM [56] and malignant testicular teratoma [57]. This suggests that while c-MYC expression is mostly cytoplasmic in cancer [48], either nuclear or cytoplasmic expression of c-MYC may contribute to tumorigenesis [48].

Tumorsphere formation assay, which is commonly used to demonstrate CSC properties in vitro, was utilized in our study [58]. This revealed that the majority of the HNmMM-derived primary cell lines cultured with tumorsphere media formulations in vitro formed non-adherent HNmMM tumorspheres, thus reflecting the potential for functional capacity of CSCs in vitro [58], reminiscent of ‘cancer neurospheres’ from human glioblastoma multiforme [59]. Further in vitro functional research including serial passaging of these HNmMM tumorspheres is needed (as this was only one passage) to substantiate our findings and assess multi-lineage (adipogenic, osteogenic, mesenchymal etc) differentiation, another hallmark of cells with stem cell properties [16].

The demonstration of an OCT4+/SOX2+/KLF4+/c-MYC+ CSC subpopulation within the TNs and another within the PTS of HNmMM in this study is novel. This is important as the two putative CSC subpopulations identified are both OCT4+/SOX2+/KLF4+/c-MYC+ and therefore may indicate an interaction between the TNs and the PTS in promoting tumor growth [60], although these two subpopulations are distinct from one another, each playing a role in tumorigenesis. However, this is beyond the scope of this study. Future studies with further in vitro and in vivo functional work are needed to conclusively determine the characteristics of these CSC subpopulations. While we observed a differential expression of iPSC markers in the TN and the PTS of HNmMM tissue across IHC staining and ISH, we were unable to quantify this variability using RT-qPCR as whole tissue lysis required for total RNA extraction meant sample homogenization occurred, thus abolishing this variability. However, the use of a range of different methodologies allowed us to further demonstrate heterogeneity within HNmMM.

Like normal tissues, tumors consist of heterogeneous cell populations that vary due to their apparent state of differentiation [61]. This phenotypic and functional diversity exhibited by CSC subpopulations has been noted in not only the same type of human cancer but also in different subpopulations of CSCs that vary from patient to patient [46], thus giving rise to the idea of intratumoral heterogeneity [62]. This idea of heterogeneity within CSCs extends beyond tumorigenesis to comprise epigenetic, local environmental, and genetic differences that could have implications for targeted therapy [63]. This concept is particularly evident in our study where the abundance of iPSC markers within HNmMM is variable using different methods. Comparing expression of these markers in HNmMM tissue, we find that the level of expression of NANOG was the least abundant in IHC staining (Figure 2) compared to the relatively high level of expression in ISH (Figure 5). SOX2, on the other hand, was the most abundant on IHC staining (Figure 2) with low expression levels detected by ISH (Figure 5) and RT-qPCR (Figure 6A) in the tissue samples. In contrast, the expression of c-MYC was abundant in HNmMM tissue across IHC staining (Figure 2), ISH (Figure 5), and RT-qPCR (Figure 6A) analyses. As all the samples were acquired from a single tumor type, MM, with loco-regional metastasis to a particular region in the body (the head and neck lymph nodes), one might expect to see similar expression levels of the markers across different methodologies of detection. However, the statistically significant differences between the iPSC-related factors studied allude to the presence of a potential intratumoral heterogeneity within HNmMM. This is consistent with reports of intratumoral heterogeneity of other melanoma markers observed in animal and cell models of MM [62].

Another possible explanation for the variability seen in the results from ISH and RT-qPCR compared to the IHC analysis is that the HNmMM-derived cells cultured for 4-5 passages in the former may not closely resemble the cells in the tissue directly biopsied from the HNmMM patients in the latter. Having said that, we also demonstrate heterogeneity of the CSCs with the expression of NANOG in only two out of the 20 HNmMM tissue samples, while KLF4, shown to occupy the NANOG promoter and thus regulate its expression [64], is present within all 20 of the HNmMM samples. Experiments show that in ESCs, the cells remain undifferentiated upon KLF4 knockdown as NANOG takes over, but knockdown of NANOG, even in the presence of KLF4, leads to cell differentiation suggesting the upstream regulation NANOG by KLF4 [64]. Taken together, it is likely that our results reflect the hierarchical nature of CSCs within HNmMM.

Overall, our findings demonstrate the presence of two putative subpopulations of CSCs expressing iPSC markers within the TNs and PTS of HNmMM. This enables us to speculate the plastic nature of the CSC hierarchy, where progenitor subpopulations within a tumorigenic niche may be induced to re-acquire a primitive pluripotent CSC state, with the expression of these iPSC-related factors, which are better characterized via protein expression. HNmMM demonstrates intratumoral heterogeneity as evidenced by the variable expression of iPSC markers across the detection methods. The exact mechanism by which CSCs sustain tumorigenesis in HNmMM is yet to be conclusively ascertained. However, the novel finding of CSC subpopulations in HNmMM shows that the loco-regional metastatic cells may be more stem-like than previously thought thus acting as a steppingstone for future research to better understand the regulatory pathways for CSCs that may underscore targeted therapy and thus improved outcomes for patients with HNmMM.

## Figures and Tables

**Figure 1 cells-09-00324-f001:**
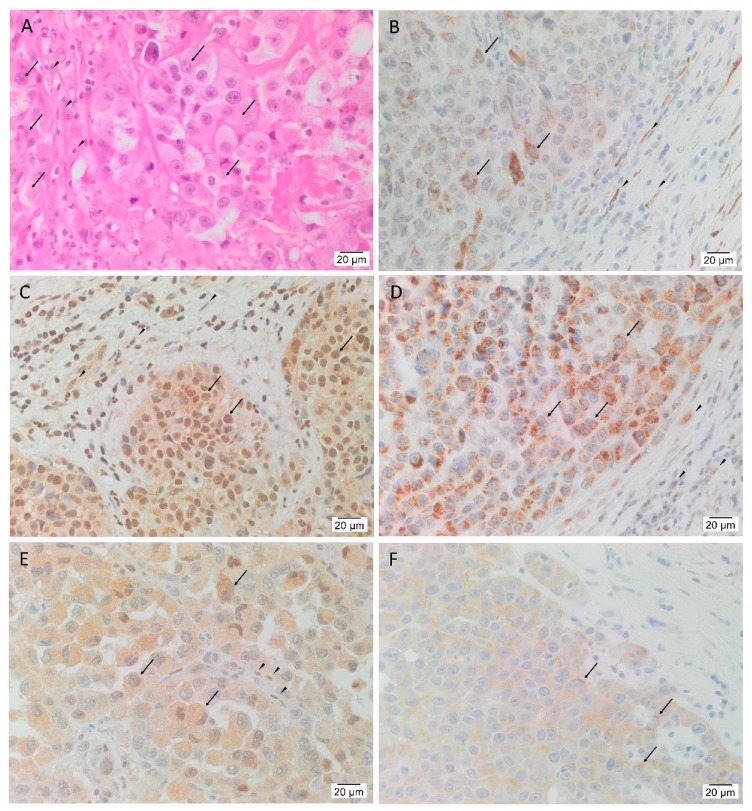
Expression of OCT4, SOX2, KLF4, c-MYC, and NANOG in head and neck metastatic malignant melanoma (HNmMM). Representative hematoxylin and eosin stained section (**A**) and immunohistochemical stained sections (**B**–**F**) of HNmMM demonstrating the tumor organized into tumor nests (TNs, arrows) surrounded by the peritumoral stroma (PTS, arrowheads). OCT4 (**B**, brown) was expressed on the nucleus and cytoplasm of the cells within the TNs and the cells within the PTS. SOX2 (**C**, brown) was predominantly expressed in the nucleus of the cells within the TNs with some cytoplasmic expression in the cells within the TNs and the cells within the PTS. KLF4 (**D**, brown) was expressed predominantly in the cytoplasm of the cells within the TNs with weak staining in the cells within the PTS. Moderate cytoplasmic and focal nuclear expression of c-MYC (**E**, brown) was present in the cells within the TNs and the cells within the PTS. NANOG (**F**, brown) was expressed in the cytoplasm of the cells within the TNs in two of the 20 tissue samples. Nuclei were counterstained with hematoxylin (**B**–**F**, blue). Original magnification 400×; *n* = 20.

**Figure 2 cells-09-00324-f002:**
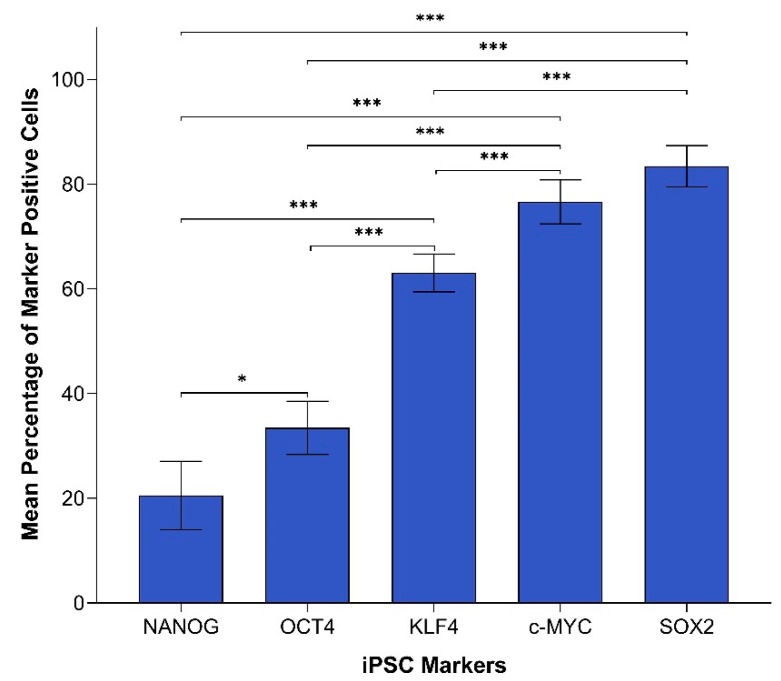
Graph demonstrating mean percentage positive expression of induced pluripotent stem cell markers NANOG, OCT4, KLF4, c-MYC, and SOX2 by cells within the tumor nests and the peritumoral stroma on immunohistochemical sections of head and neck metastatic malignant melanoma. Error bars represent 95% confidence intervals of the mean. Three replicates from each of the 19 patient tissue samples were used for an Analysis of Variance (ANOVA), thus giving a sample size of 57 for each of the following markers: OCTs, SOX2, KLF4, and c-MYC (*n* = 57). Similarly, for NANOG, three replicates from each of the two patient tissue samples were used for an ANOVA, thus giving a sample size of 6 (*n* = 6). ***, *p* < 0.0005; *, *p* < 0.05.

**Figure 3 cells-09-00324-f003:**
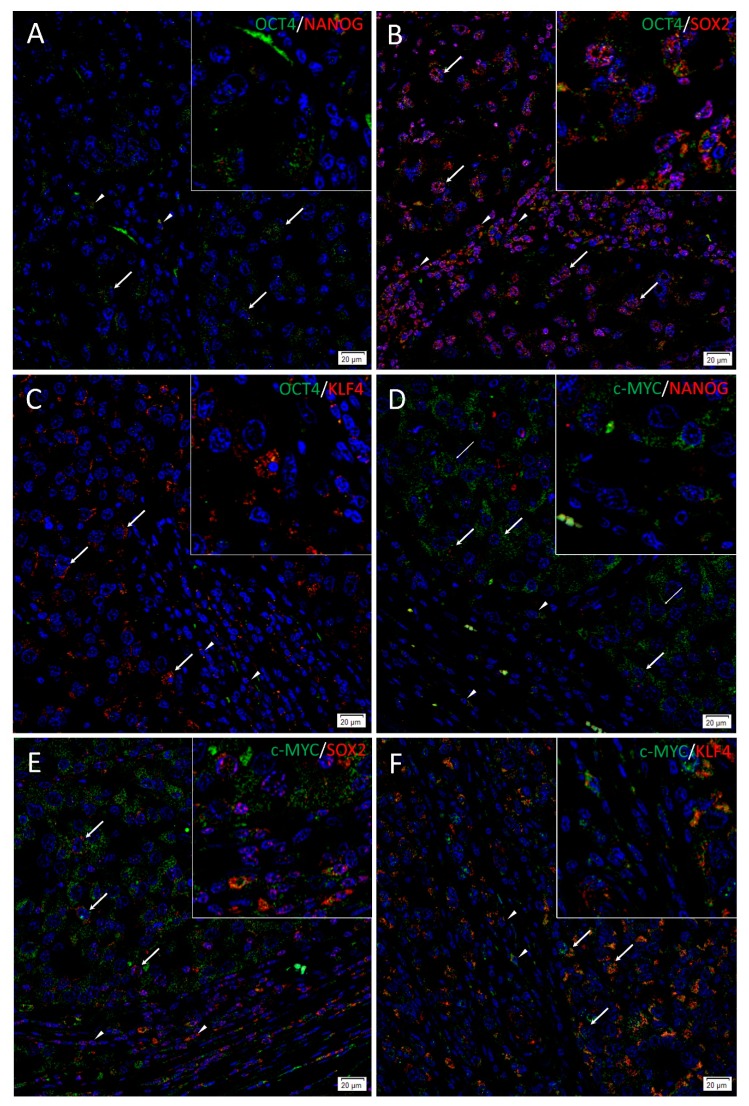
Immunofluorescence stains of head and neck metastatic malignant melanoma tissue samples demonstrating expression of induced pluripotent stem cell markers. Representative sections showing expression of OCT4 (**A**–**C**, green), SOX2 (**B**, red), and KLF4 (**C**, red) were expressed on the cells within both the tumor nests (TNs, thick arrows) and the peritumoral stroma (PTS, arrowheads). The c-MYC+ [(**D**–**F**), green] cells within the TNs and the PTS also expressed SOX2 (**E**, red) and KLF (**F**, red). NANOG (**A**, red) was expressed on the OCT4+ (**A**, green) cells within both the TNs (thick arrows) and the PTS (arrowheads), and some cells (thick arrows) and not the others (thin arrows) within the TNs. All slides were counterstained with 4′,6′-diamidino-2-phenylindole (blue). Original magnification 400×; *n* = 2. The insets show enlarged views of the corresponding images.

**Figure 4 cells-09-00324-f004:**
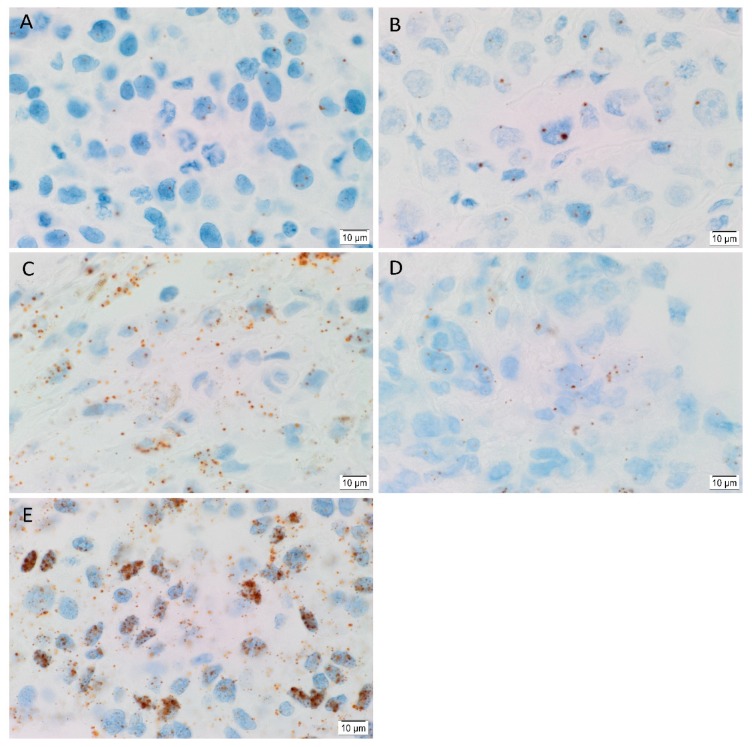
Representative in situ hybridization images of head and neck metastatic malignant melanoma tissue samples demonstrating mRNA transcript expression (brown) of induced pluripotent stem cell markers OCT4 (**A**), NANOG (**B**), SOX2 (**C**), KLF4 (**D**), and c-MYC (**E**). Nuclei were counterstained with hematoxylin (**A**–**E**, blue). Original magnification 1000×; *n* = 6.

**Figure 5 cells-09-00324-f005:**
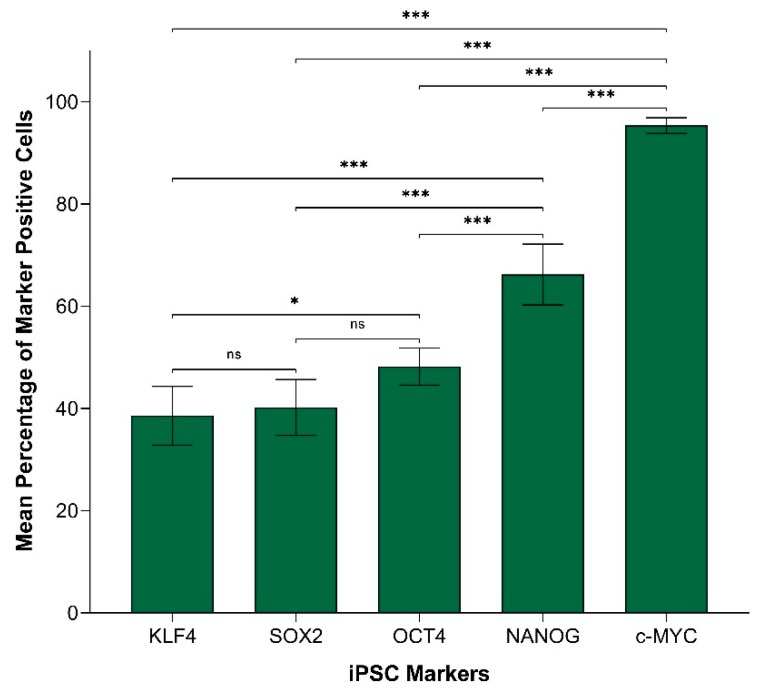
Graph demonstrating mean percentage positive expression of induced pluripotent stem cell markers KLF4, SOX2, OCT4, NANOG, and c-MYC by cells within the tumor nests and the peritumoral stroma on in situ hybridization-stained sections of head and neck metastatic malignant melanoma. Error bars represent 95% confidence intervals of the mean. Six replicates from each of the six patient tissue samples were used for ANOVA, thus giving a sample size of 36 for each marker (*n* = 36). ***, *p* < 0.0005; *, *p* < 0.05; ns, not statistically significant.

**Figure 6 cells-09-00324-f006:**
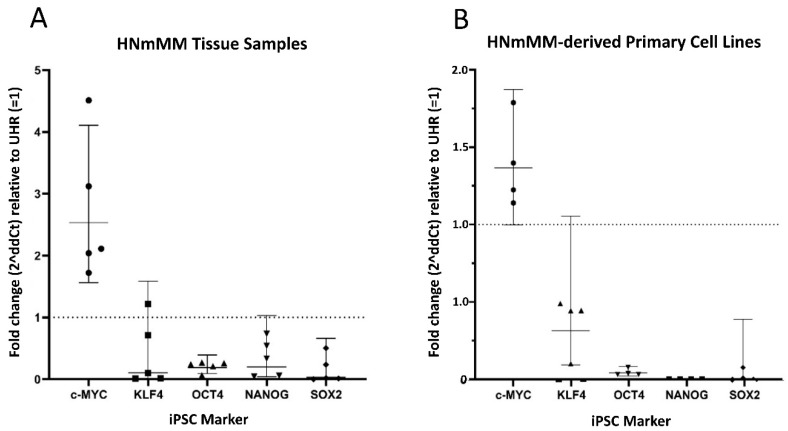
Quantitative transcriptional profiling of induced pluripotent stem cell (iPSC) markers c-MYC, KLF4, OCT4, NANOG, and SOX2. Graphs demonstrating average fold change (2^∆∆Ct^) values of triplicate RT-qPCR runs carried out on five snap-frozen head and neck metastatic malignant melanoma (HNmMM) tissue samples (**B**) four HNmMM -derived primary cell lines (**A**) amplifying mRNA transcripts for these iPSC markers. CT values of the iPSC markers were normalized to the housekeeping genes GAPDH and PSMB4 to calculate ΔCT, and expression compared relative to that of UHR (y = 1). Error bars represent 95% confidence intervals of the geometric mean.

**Figure 7 cells-09-00324-f007:**
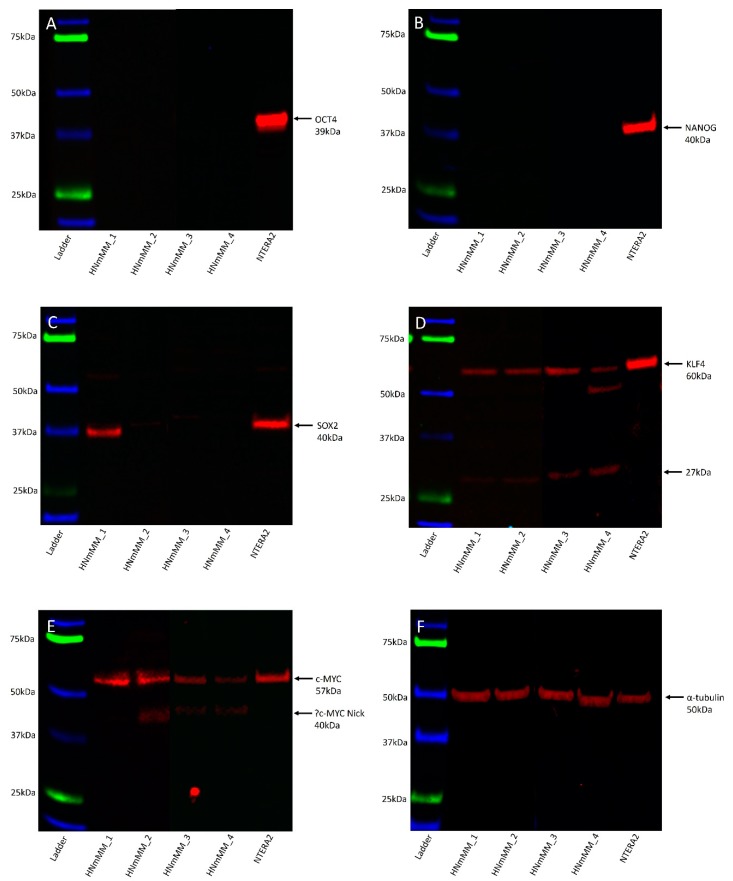
Representative Western blot images of total protein extracted from four head and neck metastatic malignant melanoma-derived primary cell lines probed for induced pluripotent stem cell markers. Blots were probed for OCT4 (**A**), NANOG (**B**), SOX2 (**C**), KLF4 (**D**), and c-MYC (**E**) and detected with HRP conjugated goat anti-rabbit antibody. α-Tubulin was used as the loading control (**F**) and detected using HRP conjugated mouse IgGĸ binding protein. Arrows indicate the presence of the proteins with their expected band sizes.

**Figure 8 cells-09-00324-f008:**
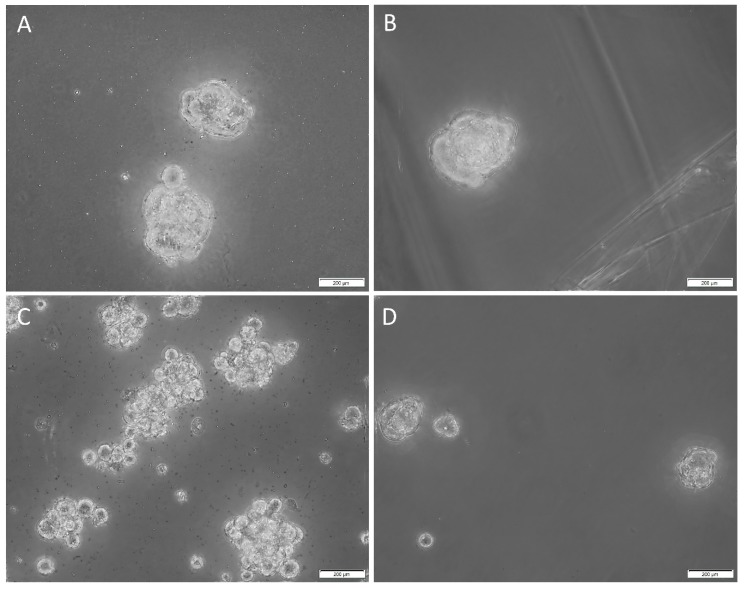
Representative high-magnification z-stack (40×) images of tumorsphere formation in suspension culture of the three head and neck metastatic malignant melanoma (HNmMM)-derived primary cell lines, averaging 62.7 µm in diameter (**A**–**C)**. A representative high magnification (40×) image of one HNmMM-derived primary cell line that did not form spheres, as defined by the tumorsphere criteria in the Methods section; instead, they averaged a diameter of 21 µm (**D**).

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
