# Peer review of "Identification of Cancer Stem Cell Subpopulations in Head and Neck Metastatic Malignant Melanoma"

_cells, 2020, doi:10.3390/cells9020324_

Round 1
Reviewer 1 Report
The authors of this manuscript focus on defining the presence of stem cell markers in HNmMM. Their work is predominantly descriptive, showing IHC, ISH, IF for OCT4, NANOG, SOX2, KLF4 and c-MYC. They support these findings with western blots. The idea that cancer stem cells remain in tumor nests is not new but important to address in this context. However, some modifications would make their study better. For instance, in Figure 1, it would help to demarcate what the authors are considering tumor nests and peritumoral stroma. Arrows and arrowheads are not sufficient for readers to appreciate these areas. Also, including a lower magnification image for each panel would also help in understanding the location of TN and PTS. The figure legend for Figure 1 describes the panels out of order. Putting them in the order that they appear in the figure is recommended. The immunofluorescence in Figure 2 is not convincing. It would help to include a melanoma marker so we can see where the tumor cells are in each panel. Supplemental Figure 2 is not bright enough be meaningful in any way. Please increase the brightness of these images (to the same extent of course) or retake them with the laser set higher so as to have clear figures. Figure 5 needs to have statistical bars showing which data bars are being compared and whether these comparisons are significant. Also, for Figure 5, how are the authors defining "positive expression?" In other words, what is the threshold for calling a signal "positive?" This needs to be defined. In figure 7, many of the labels for the lanes are not correctly lined up. In addition, it is unclear why the authors assessed sphere-forming capacity. What does the ability to form spheres mean with regard to stem cells? this is not mentioned and needs to be along with relevant references. The authors conclude that their data demonstrate that HNmMM contain subpopulations of CSCs. Their data do not show this. To make this claim, they would need to demonstrate that these cells are self-renewing and pluripotent.
Author Response
The authors of this manuscript focus on defining the presence of stem cell markers in HNmMM. Their work is predominantly descriptive, showing IHC, ISH, IF for OCT4, NANOG, SOX2, KLF4 and c-MYC. They support these findings with western blots. The idea that cancer stem cells remain in tumor nests is not new but important to address in this context. However, some modifications would make their study better. For instance, in Figure 1, it would help to demarcate what the authors are considering tumor nests and peritumoral stroma. Arrows and arrowheads are not sufficient for readers to appreciate these areas. Also, including a lower magnification image for each panel would also help in understanding the location of TN and PTS.
Authors’ Response: We are grateful for the reviewer’s comments and helpful suggestions. We have added a low magnification (200x) H&E image of a representative HNmMM tissue sample and illustrated the demarcation between the TNs and PTS as suggested. In addition, we have included similar lower magnification IHC images as a supplemental figure (Suppl. Fig. 1) to provide a better perspective.
The figure legend for Figure 1 describes the panels out of order. Putting them in the order that they appear in the figure is recommended.
Authors’ Response: We have reshuffled the images to include NANOG in the last panel.
The immunofluorescence in Figure 2 is not convincing. It would help to include a melanoma marker so we can see where the tumor cells are in each panel.
Authors’ Response: We have provided co-staining of Melan-A with iPSC markers NANOG, SOX2, KLF4 and c-MYC as a supplemental figure (Figure S3) although we are unable to carry out Melan-A/OCT4 co-staining because of primary antibody incompatibility.
Supplemental Figure 2 is not bright enough be meaningful in any way. Please increase the brightness of these images (to the same extent of course) or retake them with the laser set higher so as to have clear figures.
Authors’ Response: We have remedied this by providing larger-sized split images for this supplemental figure (now Figure S4) which spans 3 pages.
Figure 5 needs to have statistical bars showing which data bars are being compared and whether these comparisons are significant.
Authors’ Response: We have added the statistical bars as suggested and updated the figure.
Also, for Figure 5, how are the authors defining "positive expression?" In other words, what is the threshold for calling a signal "positive?" This needs to be defined.
Authors’ Response: We have added a statement on cell counting in the Methods section (Section 2.5) in our amended manuscript.
In figure 7, many of the labels for the lanes are not correctly lined up.
Authors’ Response: Thank you for picking this up. We have amended this in the updated the figure.
In addition, it is unclear why the authors assessed sphere-forming capacity. What does the ability to form spheres mean with regard to stem cells? this is not mentioned and needs to be along with relevant references.
Authors’ Response: We thank the reviewer for the comment. We have outlined the purpose of in vitro sphere formation, to study CSC properties rather than as a bio-mimicker of cancer tissues in the Results and Discussion sections of our revised manuscript and referenced “Weiswald, L.-B.; Bellet, D.; Dangles-Marie, V. Spherical cancer models in tumor biology. Neoplasia 2015, 17, 1-15, doi:10.1016/j.neo.2014.12.004.”
The authors conclude that their data demonstrate that HNmMM contain subpopulations of CSCs. Their data do not show this. To make this claim, they would need to demonstrate that these cells are self-renewing and pluripotent.
Authors’ Response: Tumorsphere formation is generally regarded as a functional assay of self-renewal property of CSCs (Weiswald et al., 2015). Cells that form tumorspheres are generally recognized as pluripotent CSCs that display the iPSC factors that imbue the tumor with self-renewal and tumorigenic properties (Lim et al., 2011; Fan et al., 2012). We feel that our IF data suggests the putative presence of subpopulations of CSCs within HNmMM tissues. However, we agree that further work such as investigating expression of pluripotency markers by these tumorspheres are needed to conclusively confirm this. We have emphasised this in the Discussion section of our revised manuscript.
accurate and reproducible comparison of gene expression data using RT-qPCR (Cronin et al., 2004); Rydbirk et al., 2016). We have amended our manuscript accordingly.
Moreover, this analysis abolish the differential expression CSC markers between TN and PTS, previously observed with the other techniques. This is due to the fact that the whole tissue is lysed and total RNA extracted, thus resulting in sample homogenization. The authors should consider and discuss this point.
Authors’ Response: We thank the reviewer for raising this point and have addressed this in the Discussion of our revised manuscript.
The IF analysis reported in Fig.3 is not easy to understand. The authors should make the figure legend easier to read and I also suggest to add the name of each evaluated marker (and the correspondent colour) in each picture thus providing a more intuitive read of the figure.
Authors’ Response: We have made the recommended changes to improve readability.

Reviewer 2 Report
The authors provided evidences of the presence of cancer stem cells (CSCs) subpopulations in head and neck metastatic malignant melanoma (HNmMM). To identify CSCs in human samples (n=20) they focused on the expression of the following pluripotent stem cell (iPSC) markers: OCT4+/SOX2+/KLF4+/c-MYC+. They demonstrated the presence of CSC populations positive for these markers in both the tumor nests (TN) and peritumoral stroma (PTS).
The work is well written and the goal of the study is clear. The experiments are well conducted and in most cases adequate controls have been included.
Yet, I have two main concerns:
In Fig.6, the authors showed the results from reverse transcriptase quantitative polymerase chain reaction (RT-qPCR) analysis of HNmMM samples. Even if the main aim is to compare the expression of each marker respect to every other marker, I suggest to also perform a comparative expression analysis respect to healthy tissues (if possible). Moreover, this analysis abolish the differential expression CSC markers between TN and PTS, previously observed with the other techniques. This is due to the fact that the whole tissue is lysed and total RNA extracted, thus resulting in sample homogenization. The authors should consider and discuss this point. The IF analysis reported in Fig.3 is not easy to understand. The authors should make the figure legend easier to read and I also suggest to add the name of each evaluated marker (and the correspondent colour) in each picture thus providing a more intuitive read of the figure.Author Response
The authors provided evidences of the presence of cancer stem cells (CSCs) subpopulations in head and neck metastatic malignant melanoma (HNmMM). To identify CSCs in human samples (n=20) they focused on the expression of the following pluripotent stem cell (iPSC) markers: OCT4+/SOX2+/KLF4+/c-MYC+. They demonstrated the presence of CSC populations positive for these markers in both the tumor nests (TN) and peritumoral stroma (PTS).
The work is well written and the goal of the study is clear. The experiments are well conducted and in most cases adequate controls have been included.
Authors’ Response: We thank the reviewer for the support of our work and insightful comments.
Yet, I have two main concerns:
In Fig.6, the authors showed the results from reverse transcriptase quantitative polymerase chain reaction (RT-qPCR) analysis of HNmMM samples. Even if the main aim is to compare the expression of each marker respect to every other marker, I suggest to also perform a comparative expression analysis respect to healthy tissues (if possible).
Authors’ Response: We thank the reviewer for the comment. The tissue samples used in this study were metastatic melanoma to the parotid or cervical lymph nodes (LNs) of the patients who underwent surgery for regional metastasis from cutaneous melanoma in the head and neck area. Obtaining healthy tissue samples from the skin or the normal LNs is not ethically feasible and the tissues may not necessary act as appropriate controls. In the absence of an appropriate normal control tissue, we have repeated the RT-qPCR experiments using universal human reference RNA (UHR) as a control for the ∆∆Ct analysis, and presented transcript expression of the iPSc markers as a fold change relative to UHR. The qPCR UHR, a mixture of total RNAs pooled from multiple adult human tissues chosen to depict a wide range of expressed genes, is regarded as a reliable reference standard for the accurate and reproducible comparison of gene expression data using RT-qPCR (Cronin et al., 2004); Rydbirk et al., 2016). We have amended our manuscript accordingly.
Moreover, this analysis abolish the differential expression CSC markers between TN and PTS, previously observed with the other techniques. This is due to the fact that the whole tissue is lysed and total RNA extracted, thus resulting in sample homogenization. The authors should consider and discuss this point.
Authors’ Response: We thank the reviewer for raising this point and have addressed this in the Discussion of our revised manuscript.
The IF analysis reported in Fig.3 is not easy to understand. The authors should make the figure legend easier to read and I also suggest to add the name of each evaluated marker (and the correspondent colour) in each picture thus providing a more intuitive read of the figure.
Authors’ Response: We have made the recommended changes to improve readability.
Reviewer 3 Report
The paper is potentially interesting in the field of CSC in Head and Neck Metastatic Malignant Melanoma.
Nevertheless, I have to raise the following concerns:
1)The authors study the expression of OCT4, NANOG, c-M;YC and KLF4, but in the analysis they did not distinguish between tumor nest and peritumoral stroma, they cound and mix togheter the results, as far as I understand. Can they provide the counting separating tumor nest and peritumoral tissues?
2) The results in Fig 2 and Fig 5, in terms of percentage of marker positive cells, show two different results: in IHC, SOX2 is the most expressed, while in in situ hybridization, c-MYC is the most expressed marker. In RNA, c-myc is the most expressed. The authors is the discussion commented on that: "A sub-group of tissue samples from the original cohort used for IHC staining was included for ISH and RT-qPCR analyses. Therefore, we speculate that the expression levels and hierarchy of markers observed in HNmMM tissue may be reflected in IHC analysis while ISH and RT-qPCR may reflect the inherent intratumoral heterogeneity"
This explanation do not sound convincing to me. Can they explain better?Have they stained consecutive slice for both techniques IF and IHC?Of course intratumoral heterogeneity can affect these kind of results but maybe staining consecutive slide of the same tumor region may help to reduce these variability in results. Have they used normal tissue as control?
3)In figure 2, Sox2 is very nuclear, but in Fig 3 is mostly cytoplasmic. Can the authors explain that?
4)The expression of the CSC markers is pretty lost upon cultures, apart from c-myc. Can they explain that?Have they compare the expression of these markers in 2D cultures versus tumorsphere culture to see whether the expression in increased in tumorsphere cultures versus 2D culture?
5)Tumorpheres formation by itself is a good readout os CSC content, but the authors should try to trypsinize the 3D culture and seed again to check for self-renewal in vitro.
6)The paper is mostly about to show the expression of potential CSC markers in Head and Neck Metastatic Malignant Melanoma. Nevertheless, if they want to prove that each of them is important in CSC, they should either over-express them of silence them to check the effect upon tumorspheres formation
7) In figure 8C, looks like there is a contamination in the culture or a lots of cell debris. Can the authors comment on that?
Round 2
Reviewer 1 Report
The manuscript is generally clear and provide evidence for CSCs in HNmMM. There are a few concerns however.
Figure 1. The figure legend does not describe the panels in the order in which they are displayed. This is very difficult to follow and confusing. If they authors put the panels in the order in which they are described in the figure legend, that would be a good fix. In addition, lower magnification images would be better (to replace the current figures or include in addition to the current figures).
A clear definition of what is meant by "tumor nest" would be helpful and including arrows or dotted line/circle to demarcate in the images what the authors are referring to as the tumor nest would be helpful.
Figure 3. The IF staining is not very convincing. Higher magnification may help. It would also help to include a melanoma tumor cell marker to define where tumor is in the stained sections.
Figure 5. Lacking statistical analysis in the figure. In addition, it is not clear how the authors defined as positive expression. What is the threshold used to define a positive signal?
Figure 7. In many of the western blots, the label for the lanes does not line up with the actual lane in the figure. Please fix.
It is unclear why the authors analyzed sphere-forming ability of the CSCs. What does this ability tell you about the cells? This needs to be clearly noted in the text otherwise, the assay makes no sense and just comes out of the blue.
On Page 15, line 446, the authors state: "....locoregional metastatic cells ARE more stem-like thane previously thought...." The authors have not proven this. Stemness is defined by self-renewing capacity and pluripotency. They did not show this. They should change "are" to "may be more stem-like"
Reviewer 2 Report
The authors answered in an exhaustive manner to all the raised issues. In the revised form the value of the study appears improved.
Author Response
We thank the reviewer for the comment and support.
Reviewer 3 Report
I Thanks the authors for their replies to my comments.
I feel satisfied to the first comment I raise:
1) The authors study the expression of OCT4, NANOG, c-M;YC and KLF4, but in the analysis they did not distinguish between tumor nest and peritumoral stroma, they cound and mix togheter the results, as far as I understand. Can they provide the counting separating tumor nest and peritumoral tissues?
Authors’ Response: We thank the reviewer for the comment. In this part of the investigation, we investigated the presence of the 5 induced pluripotent stem cell (iPSC) markers OCT4, SOX2, NANOG, KLF4 and c-MYC in head and neck metastatic malignant melanoma (HNmMM) tissue samples from 20 patients, by IHC staining. If present, we wanted to then quantify the level of expression for each of the markers, to see if there was a difference in expression between the markers. IHC staining showed that these markers were present in the HNmMM tissue samples. So we quantified their expression in each HNmMM tissue sample by cell counting to give the percentage of positively stained cells out of the total number of cells counted. It was not our aim to separately count positive cells in the tumour nests and in the peritumoral stroma for this part of the experiment. To this effect, we have found that the different iPSC markers are found in HNmMM tissue with varying expression levels within the tissue samples.
Regarding my second comments, the authors did not repply to my concerns, but they where just describing the methodologis they used in the article. In my opinion, the statement:
"A sub-group of tissue samples from the original cohort used for IHC staining was included for ISH and RT-qPCR analyses. Therefore, we speculate that the expression levels and hierarchy of markers observed in HNmMM tissue may be reflected in IHC analysis while ISH and RT-qPCR may reflect the inherent intratumoral heterogeneity" , is not supported by any observation in their work, so it should be removed from the paper.
Regarding the last part of the review, I understand now better the part relative to SOX2 staining, and I understand they do not want to make further experiments on this work, but checking by qPCR stemness genes in 2D culture versus tumorpheres will strongly support their observations.
Finally, thanks for the comment on figure 8, nevertheless the images should be improved for paper quality, 8A and 8B are blurring and the background of the 4 plots in figure 8 is totally different.
Author Response
Please see the attachement
